



# Optimal Flight Pattern Debate for Airborne Wind Energy Systems: Circular or Figure-of-eight?

Dylan Eijkelhof[1], Nicola Rossi[1,2], and Roland Schmehl[1]

[1]Faculty of Aerospace Engineering, Delft University of Technology, 2629 HS Delft, The Netherlands
[2]Università degli Studi di Trento, Via Sommarive, 9, 38123 Trento, Italy

**Correspondence:** Dylan Eijkelhof (d.eijkelhof@tudelft.nl)

**Abstract.** The computational study compares the performance of circular and figure-of-eight flight patterns for fixed-wing ground-generation airborne wind energy (AWE) systems using a PID-based basic controller that effectively controls the kite during each pattern's pumping cycle in a Matlab® Simulink® environment. A simple, adjustable control framework enables a steady analysis within consistent operational parameters, allowing for fair comparisons of power output, power quality, ground surface area requirements, and structural load impacts. The simulation results reveal that using the $150\,\mathrm{m}^2$ MegAWES reference kite at $15\,\mathrm{m\,s}^{-1}$ the circular flight pattern achieves the highest cycle-averaged power output, providing 1.85 MW at a power density of $2.94\,\mathrm{MW\,km}^{-2}$, making it advantageous for maximizing energy within limited spatial constraints. Conversely, the figure-of-eight down-loop pattern demonstrates superior power quality with lower power peaks (a peak-to-average-power ratio of 3.85) and lower expected structural fatigue due to a reduced load frequency of 0.034 Hz, supporting greater operational stability and system longevity. The up-loop variation performed the worst on all metrics considered in this work. This study offers insights into the trade-offs between energy output, efficiency, and structural demands associated with each flight path, providing a foundation for future AWE flight path selection and control strategy optimizations.

## 1   Introduction

Airborne wind energy (AWE) is an emerging technology aiming to revolutionise the renewable energy sector by harnessing the power available in higher-altitude winds. Unlike conventional wind turbines, the operating pattern is not fixed. A conventional wind turbine has rotating blades connected in the centre by a hub. AWE devices use tether kites which have a much larger potential operating domain which comes with complex controller challenges.

One shared characteristic by the different existing AWE systems, is that they all fly a closed pattern. However, the best shape of this pattern has not yet been decided on. The debate about the optimal flight pattern dates back to the beginning of the technology development in the late 2000s. The industry appears undecided on a single pattern, with circular and figure-of-eight paths both prevalent. Tether winding seems to be the key reason for the latter, but other factors could influence the choice. [add citation]

This work focuses on the development of a basic flight controller for a fixed-wing AWE system, designed in a Matlab® Simulink® environment, with the purpose of comparing different flight patterns. The controller is engineered to replicate the





entire pumping cycle of a ground-gen kite system, simulating the critical phases of power generation (traction) and its recovery (retraction). By modelling the kite's flight path and control dynamics within this virtual environment, the model can provide valuable insights into the performance and efficiency of different flight strategies.

In particular, this research explores and compares the flight patterns within the same operational framework. Three main patterns exist in AWE. These are divided into two shapes, circular motion and figure-of-eight shape flight. The latter can be flown in two ways, one where the outer curves of the figure are flown upwards (up-loop) and one which goes down at the sides (down-loop). These flight paths have been extensively studied in the context of AWE systems, with each offering distinct advantages and challenges in terms of stability, energy capture, and system complexity. The performance is studied using three criteria: average cycle power, power quality (oscillations) and projected ground surface area.

The MegAWES 3 MW fixed-wing reference kite (Eijkelhof and Schmehl, 2022) is used in conjunction with a new improved flight controller developed by Rossi (2023), which, in turn, is based on the work performed by Eijkelhof (2019); Eijkelhof and Schmehl (2022). The controller proposed by Eijkelhof and Schmehl (2022) is complex and requires many controller parameters to be tuned to have an optimal trajectory. This makes the controller not suited for a comprehensive study of the flight pattern. Therefore, a more simple, easier-to-tune controller is required.

Control and navigation of AWE systems often follow a hierarchical control structure, as commonly proposed for Ground-Gen rigid-wing systems (Rapp et al., 2019; Vermillion et al., 2021). This cascaded approach effectively breaks down the complex control problem into manageable sub-problems, allowing for optimal customisation of each control layer. The higher-level control is typically managed by a finite state machine, which supervises transitions across various flight operations, including launch, traction, retraction, and safety manoeuvres.

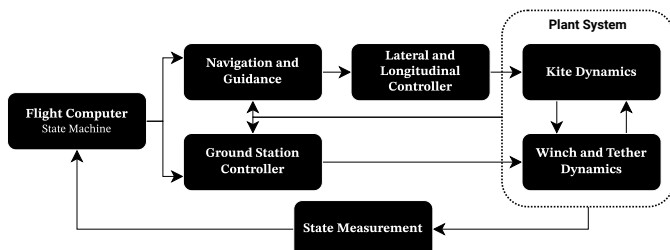

**Figure 1.** Traditional control architecture representation, featuring outer and inner control loops, for a ground-gen AWE system.

In path following, target point-based approaches are frequently utilised to guide the kite along predefined trajectories. Fagiano et al. (2014) demonstrated the feasibility of achieving periodic figure-of-eight flight paths by alternating between two target points. Fechner and Schmehl (2016) extended this approach by introducing a method that uses four-way points connected by arcs and great circles to refine the figure-of-eight pattern. Although these methods may result in less predictable path shapes, their simplicity and computational efficiency make them appealing for real-world applications.

Another method, is continuous path parametrisation, where the entire path is discretised in points which are tracked by the navigation system. A widely studied controller logic using this method is given by the $L_1$ guidance logic. This logic was initially





proposed by Park et al. (2004) and later refined by Fernandes et al. (2022) to the so-called $L_0$ guidance logic. Both methods are a widely adopted non-linear path following strategy in aerial vehicle navigation. The latter method ensures asymptotic stability and is particularly well-suited for AWE systems. It calculates the required centripetal acceleration to define a reference roll angle, facilitating precise path following and enhanced system performance. The path planning proposed in this paper is based on this $L_0$ logic.

Additionally, the control of the kite's attitude and dynamics is critical in ensuring stable and efficient AWE system operation. For flexible wings, the turn rate law proposed by Erhard and Strauch (2013) links kite steering inputs to its tangential plane course rate, allowing for precise control over the kite's flight path. However, this law does not apply to fixed-wing systems, which utilise standard actuation mechanisms like ailerons, rudder(s), and an elevator.

An example of a more developed control strategy can be seen in the Ampyx Power flight controller presented by Ruiterkamp and Sieberling (2013), where the longitudinal and lateral motions of the kite are decoupled and controlled independently. This approach involves breaking down the figure-of-eight curve into discrete target points and using linear controllers to convert these into reference roll/bank angles. However, challenges remain, particularly in maintaining tether tension within safe limits.

Other advanced control strategies which include the use of continuous path parameterisation, are presented by Jehle and Schmehl (2014) and by Rapp et al. (2019). Rapp et al. combine a PID controller for state-ideal path error correction with non-linear dynamic inversion to achieve the desired kite attitude. Lyapunov-based controllers, as developed by Li et al. (2015), focus on controlling kite attitude and rate error dynamics through differential and proportional control laws, addressing both translational and rotational motions independently. The cascaded PID control loop, used for the attitude control of the kite in this work, is based on Eijkelhof (2019), however, this is in turn a simplified version of Rapp et al. (2019).

To have a consistent system for the analysis of the flight paths, the reference system described by Eijkelhof and Schmehl (2022) is used. Several parts have been improved since the publication. A new winch design and controller is implemented as developed in Hummel (2023). The winch has been better sized for the system size of the MegAWES reference kite, and the winch controller has been changed to follow a power-optimal strategy.

In this work, we will provide a comprehensive overview of the numerical framework and describe the design and implementation of the flight controller in Matlab® Simulink® (Section 2.1), present the reference system and winch controller in (Section 2.4). A detailed analysis of the simulation results comparing the circular and two figure-of-eight flight paths is presented thereafter (Section 3). Finally, a conclusion is drawn based on the results (Section 4).

## 2 Methodology

In this section, the numerical framework is described, which is used to compare the three different flight paths. First, the flight controller is discussed, split up into lateral and longitudinal control and the reference systems. Second, the reference kite model and winch are discussed.



## 2.1 Flight controller

The development of the flight control system for the fixed-wing AWE system is crucial for ensuring reliable and efficient operation. This system is designed to handle the intricate dynamics of tethered flight, which involves managing both the kite's lateral and longitudinal motions. The control architecture is based on a cascaded control approach, which separates the control tasks into more manageable sub-problems, allowing for precise tuning and optimisation of each component. This section outlines the methodologies employed in designing the state machine for switching between phases, the lateral and longitudinal control systems of the kite, as well as the reference systems used to model the kite's flight dynamics accurately.

Figure 2 illustrates the control state machine structure designed for the system, detailing the transitions and control objectives for each flight phase. This state machine framework enables smooth transitions between traction, retraction, and other phases, optimizing power output and stability under varying operational conditions.

In the traction phase, lateral and pitch control are focused on maintaining the maximum angle of attack for optimal energy capture, while winch control enforces an optimal force set point to manage tether tension. The transition phase from retraction to traction adjusts lateral and pitch controls similarly to the traction phase, driving the angle of attack to its peak to ensure longitudinal stability until the kite reaches the designated pattern target point to go back into traction. During the retraction phase, lateral control reorients to a pre-determined retraction target point, pitch control tracks a descent angle, and winch control reduces the tether tension to a reference force set point, effectively preparing the system for the next cycle.

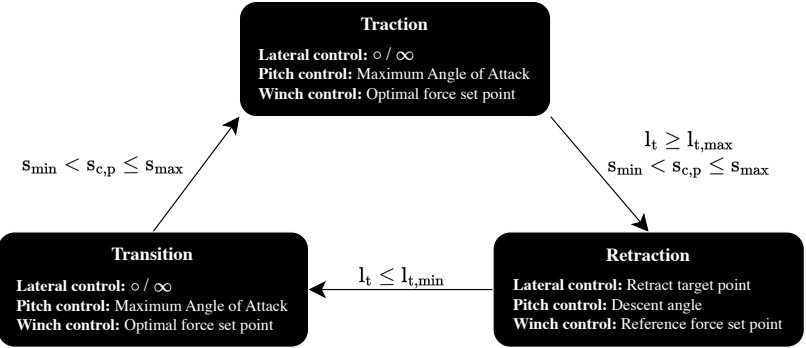

**Figure 2.** The control state machine describing the transitions and control objectives for each flight phase.

### 2.1.1 Lateral control

The lateral controller requires a controller objective. This objective is determined by the path following module. This work utilises an adapted version of the $L_1$ controller logic, a discretised $L_0$ controller logic. As both methods are similar in principle, the $L_1$ method is described first, from which the $L_0$ is then derived.

The $L_1$ controller logic is typically a control strategy used in guidance systems in flight controllers for path following, where a dynamic system must track a desired trajectory. The key principle behind the $L_1$ logic is that it defines a distance metric, often referred to as the $L_1$ distance, which measures the separation between the actual position of the system (e.g. kite or





aircraft) and a designated target point on the intended path. This distance is often used as a control parameter to ensure that the system moves toward the desired trajectory. The $L_1$ controller has only one adjustable design parameter, the $L_1$ distance, which can be tuned to achieve faster or smoother convergence to the desired path. This makes the strategy simple to tune and ideal in the context of this work. Park et al. (2004) demonstrates the effectiveness of this tracking method and explains how to optimise the parameter across different scenarios, including straight lines, perturbed paths, and circular trajectories. The

approach guarantees asymptotic stability, as shown by a Lyapunov invariant set theorem analysis. This makes the $L_1$ control strategy particularly well-suited for non-linear systems, offering an advantage over traditional linear control methods.

Rather than specifying the $L_1$ distance as a control metric, a discretized interpretation of the $L_0$ distance is used (Fernandes et al., 2022). In this approach, the distance to the target point is calculated from the nearest point along the trajectory projected from the kite's position.

In contrast to conventional methods, this formulation broadens the field of attraction toward the desired course, guaranteeing that the guidance system consistently retains a target point on the path, even when the kite drifts further from the desired trajectory. When the $L_1$ distance is used, the controller becomes confused when the kite is further away from the trajectory than the specified $L_1$ distance. There are several ways to solve this, for example, by widening the distance when it cannot converge on a point on the trajectory. However, using the $L_0$ distance instead deemed to be the most effective.

Due to the closed and intersecting geometry of certain flight paths, accurately selecting the target point becomes more complex. When dealing with a continuous curve, iterative techniques, such as Newton-Euler methods, are typically employed to determine both the nearest and target points with a manageable number of iterations. However, for the discrete representation of the path as a finite set of points, an alternative approach is necessary.

In the studied scenario, the set of path points is discretised using a variable $s$ over the interval from 0 to $2\pi$. This ensures

unique tracking points, independent of path shape. The path is discretised on the tangent plane and then projected onto a spherical surface. Using a path parametrisation dependent on this variable $s$, Cartesian path coordinates can be calculated and stored.

The discretisation of the circular path is straightforward given the definition of a circle. The figure-of-eight can be defined in several different ways, a Lissajous or (hyperbolic) Lemniscate of Booth, for example. The latter is used in this framework

and given the previously mentioned parameter $s$ is defined in the tangent plane by Equations (1) and (2).

$$x_{\text{fig8}}(s) = \frac{1}{h_\tau} \frac{b_{\text{Booth}} \sin(s)}{1 + \left(\frac{a_{\text{Booth}}}{b_{\text{Booth}}}\right)^2 * \cos(s)^2}, \tag{1}$$

$$y_{\text{fig8}}(s) = \frac{1}{h_\tau} \frac{a_{\text{Booth}} \sin(s) \cos(s)}{1 + \left(\frac{a_{\text{Booth}}}{b_{\text{Booth}}}\right)^2 * \cos(s)^2}, \tag{2}$$

where $a_{\text{Booth}}$ and $b_{\text{Booth}}$ define the height and width of the figure of eight, respectively, and $h_\tau$ is the distance between the tangent plane and the ground station. Even though the number of path points could be freely chosen, for this study 50 points

were selected for $s$. This leads to a smooth path discretisation but is still computationally efficient.



Additionally, a distance vector is introduced, containing the distances between the kite's current position and each point on the path. This vector is organised cyclically, allowing the algorithm to continually search for the closest point as the kite follows the closed-loop trajectory. The index in the $s$ vector corresponding to the closest point aligns with the index of the minimum value in the distance vector. To select the correct nearest point, special attention is needed for complex paths like the figure-of-eight. In this shape, the kite crosses over multiple intersections, which can lead to wrong tracking points. The overcome this problem, the selection process uses a smaller subset of the distance vector, starting from the previous time step. This proved to be a reliable method to choose the correct projected kite location on the trajectory.

Instead of choosing a distance value as the controller parameter, the tracking point is determined using a set number of elements ahead. After identifying the closest point, the target point is set as the $n^{\text{th}}$ element ahead in the sequence. This guarantees that the target point remains consistently defined as $n$ points ahead of the nearest one, where $n$ serves as a tunable discrete control parameter.

The lateral controller determines the actuator demands, based on the tracking point given by the path following module. The target vector is computed using the coordinates of both the kite and the target point, within the wind reference frame described in Section 2.1.3. Using the appropriate transformation matrices, the target vector and the kite's velocity, are projected onto the tangent plane. This tangent plane is visible in Figure 6 and is tangent to the sphere surface, with a radius equal to the straight tether length and the centre at the ground station.

The resulting projections are then used to derive the heading angle $\eta$. To avoid impractically sharp turns, which could lead to hazardous manoeuvres, the calculated heading angle $\eta$ is restricted to $\pm\frac{\pi}{4}$. Finally, by considering all forces acting on the kite, the desired roll angle $\phi_{\tau,\text{des}}$, relative to the tangent plane, is determined by Equation (3).

$$\phi_{\tau,des} = \arcsin\left(\frac{a_c\, m + F_{g,y}\cos(\phi_{\tau,\text{act}}) + F_{g,z}\sin(\phi_{\tau,\text{act}})}{F_L}\right), \tag{3}$$

where $a_c$ is the lateral acceleration needed for the vehicle to trace a curved trajectory, $m$ is kite's mass, $F_g$ is the gravitational force, $\phi_{\tau,\text{act}}$ is the actual roll angle and $F_L$ is the lift force. $a_c$ can be calculated using Equation (4).

$$a_c = 2\frac{V_k^2}{L_1}\sin(\eta), \tag{4}$$

where $V_k$ is the kite velocity, and $\eta$ is the angle between the kite velocity and the direction to the reference point. The concept is visualised in Figure 3.

This desired roll angle is then converted into a demand for the aileron deflection using multiple PI controllers, illustrated in Figure 4.

### 2.1.2 Longitudinal control

One of the key factors influencing power extraction from the wind during crosswind flight of AWE systems is the tether force, which is directly linked to the kite's lift. To maintain proper tether tension, the kite's angle of attack must be controlled via the longitudinal or pitch controller.



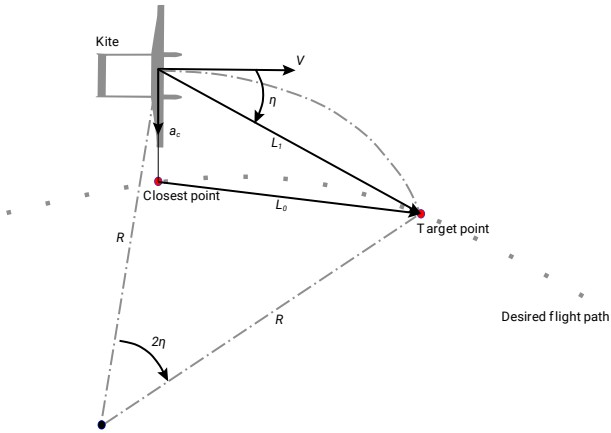

**Figure 3.** Principle of $L_1$ and $L_0$ guidance logic. The closest point to the kite and the target point are highlighted in red. $V$ is the kite velocity, $a_c$ the required acceleration and $\eta$ the heading.

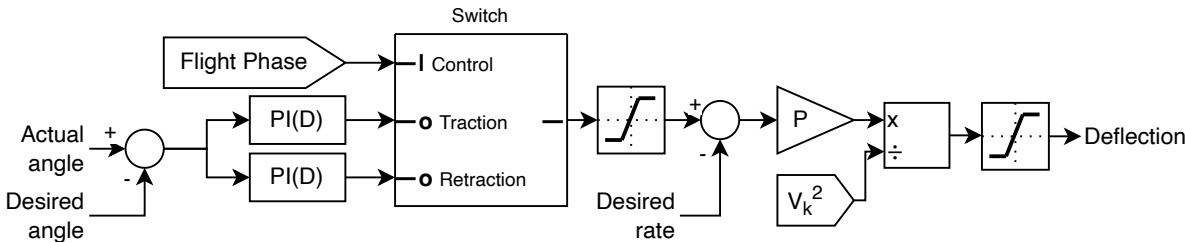

**Figure 4.** Flow diagram of the attitude and rate controller for both lateral and longitudinal control. The lateral controller needs a PID to accurately follow the desired input, and for the longitudinal controller, a PI controller suffices. The gains are tuned separately for traction and retraction. The integral part of the PI(D) controller is reset at the start of the specific flight phase to prevent windup.

From the aerodynamic model, the lift and drag forces generated by the airfoil are governed by the aerodynamic coefficients for lift ($C_L$) and drag ($C_D$), which are functions of the angle of attack ($\alpha_a$). Typically, $C_L$ increases with $\alpha_a$ until the excessive separation of airflow behind the wing causes the lift to drop.

Since the system's primary goal is to maximise wind energy extraction, the control of longitudinal flight dynamics aims to sustain a high lift-to-drag ratio to optimise power generation. Considering tether drag, the highest feasible lift-to-drag ratio is practically lower than for untethered aircraft. Hence, it is essential to consider the total system's combined lift-to-drag ratio, including the kite and tether drag. In general, a higher lift-to-drag ratio corresponds to a higher angle of attack, yielding more power. Hence, a simple approach is used, choosing a constant high angle of attack as a reference target during the power

production phase.





As developing a new reference kite and aerodynamic model is beyond the scope of this paper, the existing aerodynamic data of the MegAWES kite is used (Eijkelhof and Schmehl, 2022). Using this, a target angle of attack of approximately $4°$ was chosen for the traction phase to keep the kite operating within the linear region. Hence, the stall region is entirely avoided. While suitable for experimental purposes, for practical applications, it is critical to ensure that tether tension does not exceed
limits during winch controller saturation, which may require changing the desired angle of attack to lower the aerodynamic performance.

The desired angle of attack $\alpha_{\mathrm{des}}$ is then used as an objective for the pitch controller, which adjusts the kite's pitch by means of elevator deflection $\delta_e$. This conversion is done by the PI cascaded loops visible in Figure 4.

While following circular or figure-eight paths, the longitudinal controller must compensate for external disturbances, includ-
185 ing winch and tether forces and kite dynamics. Wind plays a significant role, particularly as the relative wind direction changes during ascent and descent. When the kite flies against the wind, the relative wind speed increases, raising $\alpha_a$, while descending in the same direction as the wind decreases $\alpha_a$. These oscillations, dependent on the velocity along the path, must be actively managed.

### 2.1.3 Coordinate frames

Various coordinate frames are used to model the entire system and effectively represent the many vector quantities that are involved. In most cases, these coordinate frames align with those commonly used in the flight mechanics literature, but some others are specially introduced for the AWE domain. This section presents a description and graphical representation of the reference frames used throughout this thesis.

In Figure 5 the earth and wind reference frames are presented. The wind reference frame has the x-axis point aligned with
195 the wind direction, which is previously defined in the earth reference system. In this study, the earth reference frame is defined as a right-handed system with the ground station placed in the origin and the z-axis pointing into the ground. This definition is coherent with the usual one defined in the aerospace literature which is also implemented in the aerospace Simulink block of the simulation framework.





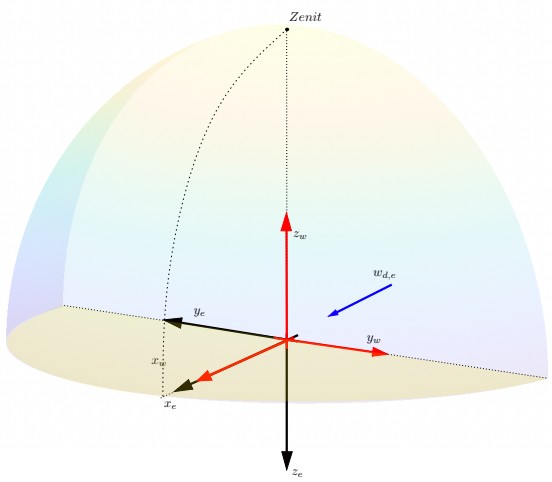

**Figure 5.** Earth and wind reference frames.

To guide the kite through the wind window and to define the desired flight path for the traction phase it is convenient to
define a reference frame centred in the kite position and tangential to the flight sphere. This frame, shown in Figure 6, can be
obtained through a sequence of transformation matrices starting from the wind coordinate system. More to the point, first a
rotation of $\theta$ around $z_w$, followed by a rotation around the new $y'$ of elevation angle $\phi$ and a translation along $x''$ of a distance
equal to the radial position of the kite. Finally, by adjusting the direction of the axes the North-East-Down (NED) coordinate
system is obtained corresponding to $x_\tau$, $y_\tau$, $z_\tau$ respectively.

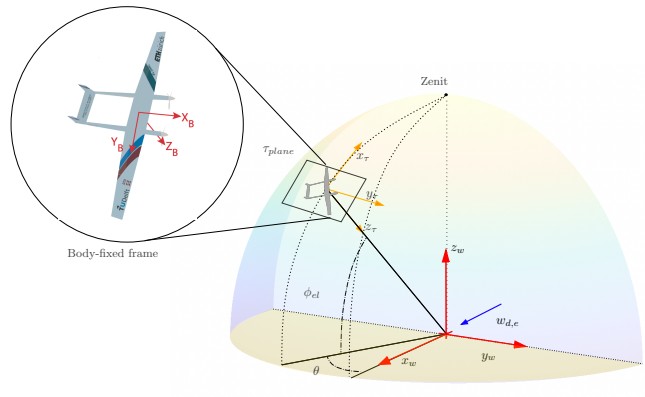

**Figure 6.** Wind and $\tau$ reference systems, visualisation of azimuthal $\theta$ and elevation $\phi$ angle and tangent plane to the wind sphere in the kite
position. On the left, a detail shows the body-fixed coordinate system.

Figure 6 also shows the body coordinate system, which is fixed in both the origin and the orientation of the kite. In this study,
the kite is considered rigid, and any deflection or deformation is not taken into account. In particular, the body-fixed frame is





centred and oriented along with the main wing, so that the $x_B$ axis points through the nose of the aircraft, the $y_B$ axis points to the right side of the main wing perpendicular to $x_B$ and the $z_B$ points down through the bottom side of the aircraft orthogonal to the $x - y$ plane.

## 2.2 Winch controller

The winch, which includes the drum, drivetrain, and generator, is modelled as a single rotating body with inertia $J$ and a constant radius $r$ (Hummel et al., 2024). The modelling approach simplifies the equations by ignoring viscous friction, as the friction torque is much smaller than the forces exerted by the tether and control torque. Additionally, the radius is assumed constant, despite the spooling of the tether, because the changes in radius are small and slow, making their dynamic impact negligible. Two primary moments act on the winch: the tether force multiplied by the winch radius and the control torque, both of which are used in the control strategy.

An "optimal" control curve is derived to define a control target. The tether force is calculated using a quasi-steady model given in Equation (5), assuming wind speed is parallel to the tether while ignoring cosine losses and the kite's mass.

$$F_t = \mathscr{C}(v_w - v_r)^2, \tag{5}$$

where

$$\mathscr{C} = \frac{1}{2}\rho S C_L E_{eq}^2 \left(1 + \frac{1}{E_{eq}}\right)^{\frac{3}{2}}, \tag{6}$$

where $F_t$ is the tether force, $v_w$ the equivalent wind speed, $v_r$ the reel-out speed, $\rho$ the air density, $S$ the reference wing area, $C_L$ the lift coefficient, $E_{eq}$ the equivalent lift-to-drag ratio (including tether drag).

Instead of modelling individual forces like gravity or wind speed variation, an equivalent wind speed is used to account for these effects. Choosing an optimal reel-out factor $f*$, to be 1/3 (Loyd, 1980), the control torque on the winch aims to balance out the torque from the tether force at steady-state. This results in a winch control curve, often called the "optimal force-squared speed manifold", where the control torque $\tau$ is the product of the optimal tether force and winch radius as shown in Equation (7).

$$\tau = 4\mathscr{C}v_r^2 r. \tag{7}$$

Hummel et al. (2024) sized the winch for the MegAWES kite as shown in Table 2. The given size parameters ensure a robust and steady response of the winch when trying to stay on the optimal control curve shown in Figure 7.

## 2.3 Optimisation scheme and objective

To support the comparative analysis of circular and figure-of-eight flight patterns, this study employs the Covariance Matrix Adaptation Evolution Strategy (CMA-ES) Auger and Hansen (2012). The CMA-ES method iteratively refines parameters by minimizing a specified objective function, allowing for targeted adjustments in both controller settings and flight path characteristics. However, rather than aiming for a globally optimal configuration, the optimization serves as a strategic exploration of



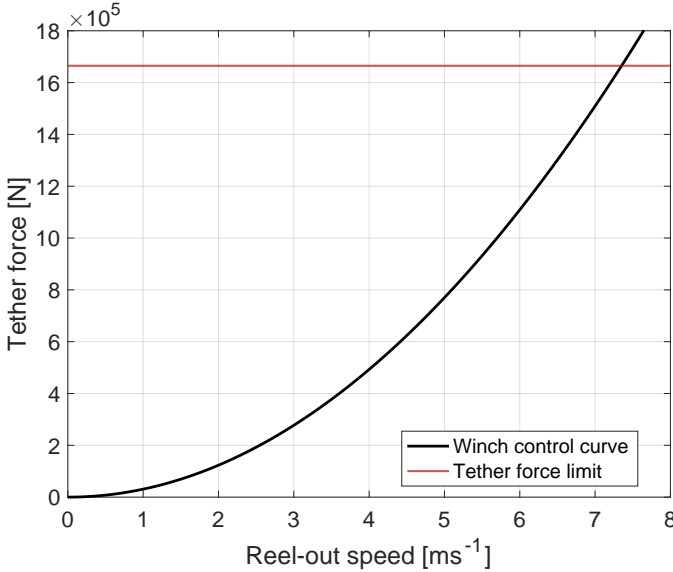

**Figure 7.** Winch control curve, showing the optimal (quasi-steady) tether force as a function of reel-out speed for an optimal reel-out factor of $\frac{1}{3}$. The MegAWES parameters are used, with $S = 150.45$, $C_L = 1.8$ and $E_{eq} = 6.08$.

the design space, focusing on identifying representative configurations for each flight pattern. This sub-optimisation approach thus provides a balanced set of configurations that enable a fair and insightful comparison of the performance, stability, and structural implications of different flight patterns, aligning directly with the study's goal of understanding each pattern's relative advantages and limitations.

Equation (8) shows the objective used with applied penalties based on exceeded prescribed maxima. Table 1 provides a set of system parameters which could be varied by the optimiser.

$$C_{\text{obj}} = -P_{\text{avg}} + p_{\text{cte}} + p_\alpha + p_{V_k} + p_{F_t} + p_{\text{papr}}, \tag{8}$$

with

$$p_{\text{cte}} = 10^2 \cdot \max(\max(\text{cross-track error})/75\text{m} - 1, 0)$$

$$p_{V_k} = 10^4 \cdot \max(\max(V_k)/100\,\text{m}\,\text{s}^{-1}, 0)$$

$$p_{F_t} = 10^4 \cdot \max(\max(F_t)/1664910\text{N}, 0)$$

$$p_{\text{papr}} = 10^5 \cdot \max(\max(\text{PAPR})/2.5, 0)$$

$$P_{\text{avg}} = \text{mean}(P_{\text{mech}})$$

---

[1]Depends on the chosen flight path.





**Table 1.** Operational and controller parameters varied during optimisations. $K_p$, $K_i$ and $K_d$ are the PID gains, respectively.

| Controller | Flight path | Tether |
|---|---|---|
| $K_p$ winch (retraction) | Fig8 $a_{\text{Booth}}$[1] | Tether length max. |
| $K_i$ winch (retraction) | Fig8 $b_{\text{Booth}}$[1] | Tether length min. |
| $K_p$ roll rate (traction) | Circle radius[1] | Retraction force |
| $K_p$ pitch rate (traction) | Elevation angle | |
| $K_p$ roll (traction) | | |
| $K_i$ roll (traction) | | |
| $K_d$ roll (traction) | | |
| $K_p$ pitch (traction) | | |
| $K_i$ pitch (traction) | | |
| $K_p$ roll rate (retraction) | | |
| $K_p$ pitch rate (retraction) | | |
| $K_p$ roll (retraction) | | |
| $K_i$ roll (retraction) | | |
| $K_d$ roll (retraction) | | |
| $K_p$ pitch (retraction) | | |
| $K_i$ pitch (retraction) | | |

## 2.4 System parameters

As this work focuses on comparing flight paths, it is of key importance as many factors as possible are kept the same throughout the analysis. Therefore, for each case study, the same kite, ground station, and tether are used. Similarly one wind shear model without turbulence is chosen and kept constant throughout the analysis. The 3 MW MegAWES kite is chosen as the ground-gen

fixed-wing kite (Eijkelhof and Schmehl, 2022). As the original MegAWES simulation framework used a winch which was not sized for the megawatt scale power, a better-sized winch is used instead (Hummel et al., 2024).

The kite is modelled as a rigid body (six degrees of freedom). The total sum of forces acting on the kite is composed of the aerodynamic force, the tether force and the gravitational force. The aerodynamic force, is a resultant force combining both lift, drag and side force.

The tether is modelled similarly to the original MegAWES simulation environment. The tether is modelled as quasi-steady, discretised using lumped masses which are connected by straight elements and have a stiffness based on the Dyneema® material.

Table 2 shows a summary of the specifications of the entire system. This includes kite design, winch size and tether specifications.



**Table 2.** General planform parameters of the wing, tail and fuselage and characteristics of the winch and tether, adapted from (Eijkelhof and Schmehl, 2022).

| Parameter | Value | | Unit |
|---|---|---|---|
| Inertia $J_{xx}$ | $5.7680 \times 10^5$ | | $\mathrm{kg\,m^2}$ |
| Inertia $J_{yy}$ | $0.8107 \times 10^5$ | | $\mathrm{kg\,m^2}$ |
| Inertia $J_{zz}$ | $6.5002 \times 10^5$ | | $\mathrm{kg\,m^2}$ |
| Mass | $6.8852 \times 10^3$ | | kg |
| *Wing / RevE*$_\mathrm{HC}$ | | | |
| Span | 42.47 | | m |
| Chord$_\mathrm{root}$ | 4.46 | | m |
| Chord$_\mathrm{tip}$ | 2.11 | | m |
| LE sweep | 2 | | ° |
| Aspect ratio | 12.0 | | - |
| Surface area | 150.45 | | $\mathrm{m^2}$ |
| Twist$_\mathrm{root \rightarrow tip}$ | $5 \rightarrow 0$ | | ° |
| *Horizontal tail / NACA0012* | | | |
| Span | 7.6 | | m |
| Chord | 2.8 | | m |
| *Vertical tail / NACA0012* | | | |
| Span | 3 | | m |
| Chord | 2.8 | | m |
| *Winch* | | | |
| Total inertia | 1.0 | $\times 10^4$ | m |
| Radius | 1.5 | | m |
| *Tether / Dyneema ®* | | | |
| Density | 0.97 | $\times 10^3$ | $\mathrm{kg\,m^{-3}}$ |
| Modulus | 1.16 | $\times 10^{11}$ | Pa |
| Ultimate strength | 3.6 | $\times 10^9$ | Pa |
| Cylindrical $C_D$ | 1.2 | | - |

## 2.5 Comparison criteria

To evaluate the flight patterns, the following key criteria are investigated: power oscillations (power quality), projected ground area, and cycle-averaged power.



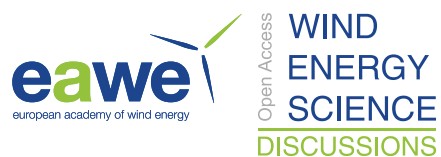

Power oscillations (power quality) reflect the stability and consistency of power generation. A system with lower power oscillations provides more stable power output, which is crucial for integration into the electrical grid. Hence, minimizing power fluctuations is a desirable characteristic in AWE systems.

The projected ground area provides an estimate of the space required for each system when deployed in a farm configuration. It is crucial to determine how dense systems can be arranged within a given surface area, directly influencing the potential energy density of the farm. Additionally, it offers a preliminary assessment of the necessary safety buffers, which are vital for mitigating the risk of system interference and ensuring operational safety.

Cycle power is the average power output over a complete pumping cycle. It serves as a primary measure of system efficiency, representing the overall energy production capability of the system. Higher cycle-averaged power indicates more efficient wind energy extraction.

## 3 Results

This section presents a detailed comparison of the circular and figure-of-eight flight patterns, evaluating their performance based on key metrics such as cycle power and peak-to-average power ratio. The simulations are performed in a consistent operational environment with a wind speed of 15 m/s across all tests. The most optimal solution is then chosen and the full pumping cycle is compared between patterns. Even though complete pumping cycles are compared, the focus has been on the traction phase where the pattern makes a big difference, the retraction strategy is relatively similar in all cases and therefore discussed in less detail.

### 3.1 Pattern optimisation

First, the cycle power is compared to the circle radius in Figure 8. As the genetic algorithm is free to choose any radius, many combinations of parameters are evaluated, hence giving many power results per radius. The optimal radius for this kite at this wind speed, given the operation conditions explained throughout this paper, is around $220 \, \mathrm{m}$. This means that the diameter of the flight pattern is around 10.5 times the wing span.

From theory, a smaller radius would lead to a larger part of the path experiencing higher wind speeds, considering cosine losses, and thus one could expect a higher power extraction. However, there is a limit on how small the circle can become before other losses and constraints become dominant. As smaller radii tend to cause higher tangential velocities, the kite velocity can exceed the prescribed maximum of $100 \, \mathrm{ms}^{-1}$. If the maximum is exceeded, the optimisation objective is penalised, steering the optimiser to lower the maximum occurring velocity. This can also be seen in the very limited amount of data points at smaller radii.

Another important metric that needs to be considered is the peak-to-average-power ratio (PAPR). The peak power of the traction phase is divided over the average cycle power. The evolution shows the penalisation of the objective is working as expected. While optimising for more cycle power, the power oscillations are minimised. In the given conditions the most optimal PAPR is a little over 4.





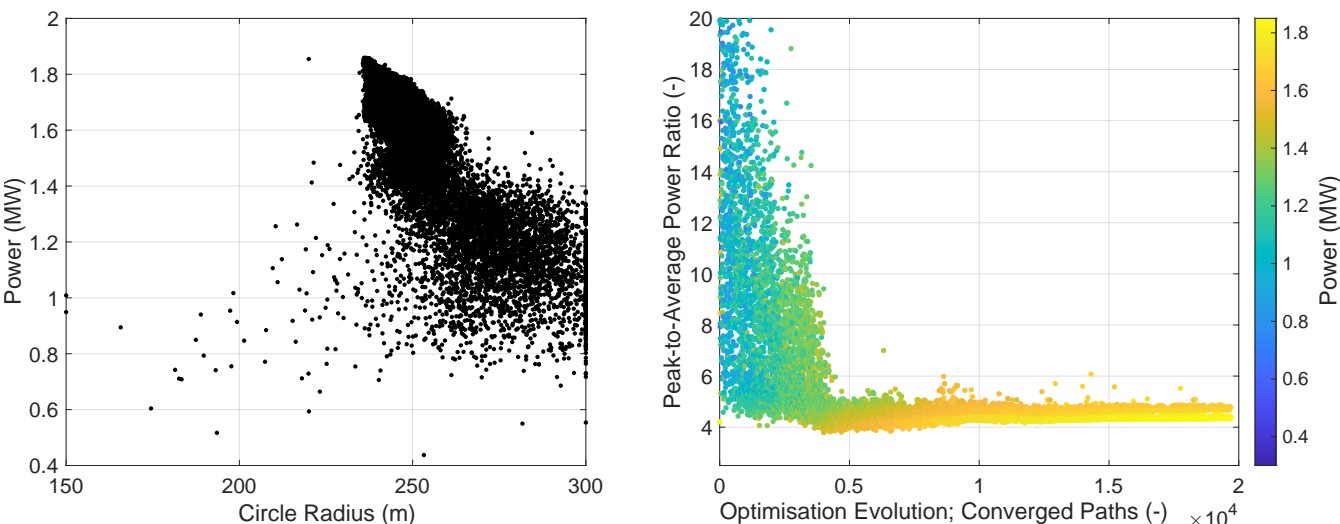

**Figure 8.** Circular pattern, 15 m/s wind speed. Left is the power with respect to different circle radii. Right is the evolution of the peak-to-average-power ratio with respect to optimisation generation.

A similar analysis is performed for both variations of the figure-of-eight. Figures 9 and 10 show the optimisation results, illustrating the geometry (width and height) of the figure-of-eight with respect to the cycle power on the left and the PAPR evolution on the right.

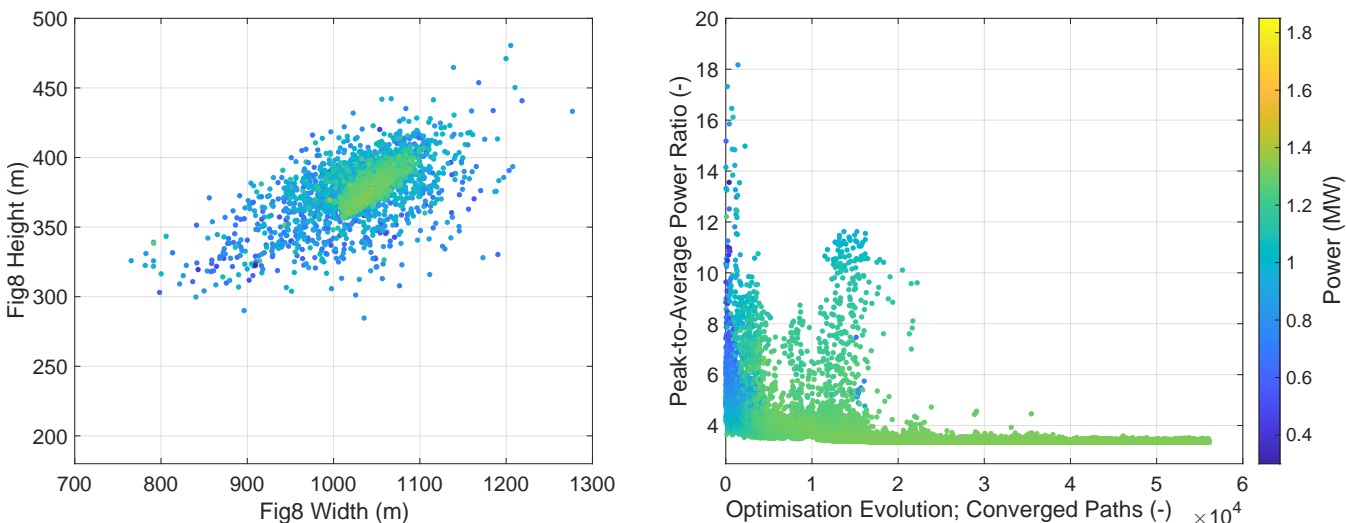

**Figure 9.** Figure-of-eight down-loop cycle power with respect to several different pattern sizes at minimum tether length, 15 m/s wind speed.




Both, down-loop and up-loop flight patterns converge to an optimal combination of width and height. The up-loop converges to smaller-sized figure-of-eights compared to the down-loop. This is the expected behaviour since especially the smaller-width
figure-of-eight patterns remain more in the effective wind window. This benefits the upward flight.

The right image in Figure 9 shows that the most optimal solution of the down-loop is limited at a lower cycle power than the circle pattern. The PAPR however, converges to the lower value of below 4.

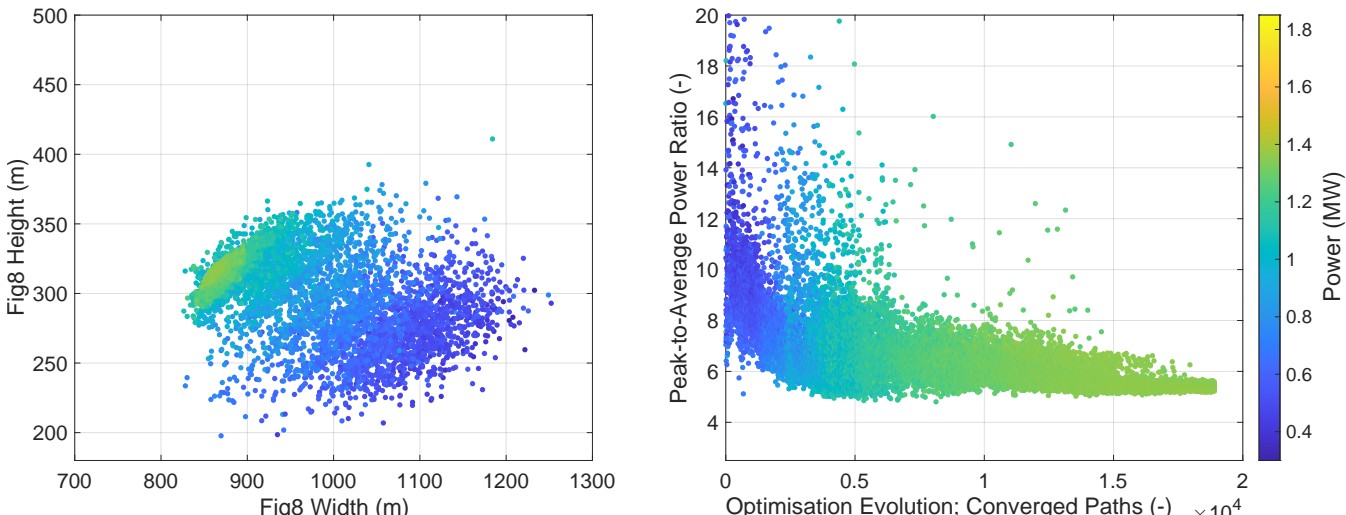

**Figure 10.** Figure-of-eight up-loop cycle power with respect to several different pattern sizes at minimum tether length, 15 m/s wind speed.

The right image in Figure 9 shows that the most optimal solution of the up-loop is similarly limited at a lower cycle power than the circle pattern. Unlike the down-loop, the up-loop PAPR however, converges to the highest PAPR value of the three
patterns.

### 3.2  Pumping Cycle Comparison

From the optimisation results, the most optimal one is selected for more detailed comparison. Here the flight patterns are compared based on the power oscillations (power quality), projected ground surface area and cycle average power.

Figure 11 shows the circular 3-dimensional flight path coloured by the mechanical power, which is shown with respect to
time on the right. The projected ground area is shown in the red dashed box on the ground plane. The centerline of the cycle trajectory is slightly shifted on the y-axis, aligning the upward flight more with the centre of the wind window. As the target path is centred at zero azimuth, this result is solely a cause of precise gain tuning and changing tracking accuracy along the pattern. Tracking accuracy can change as different velocities occur throughout the pattern making it easier and harder to track a given target.

It can be noted that at this wind speed, most of the mechanical power is positive; however, during each loop, the winch needs to retract a bit to insert energy into the system to get it to the top of the circle. This extra required energy is due to the

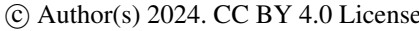



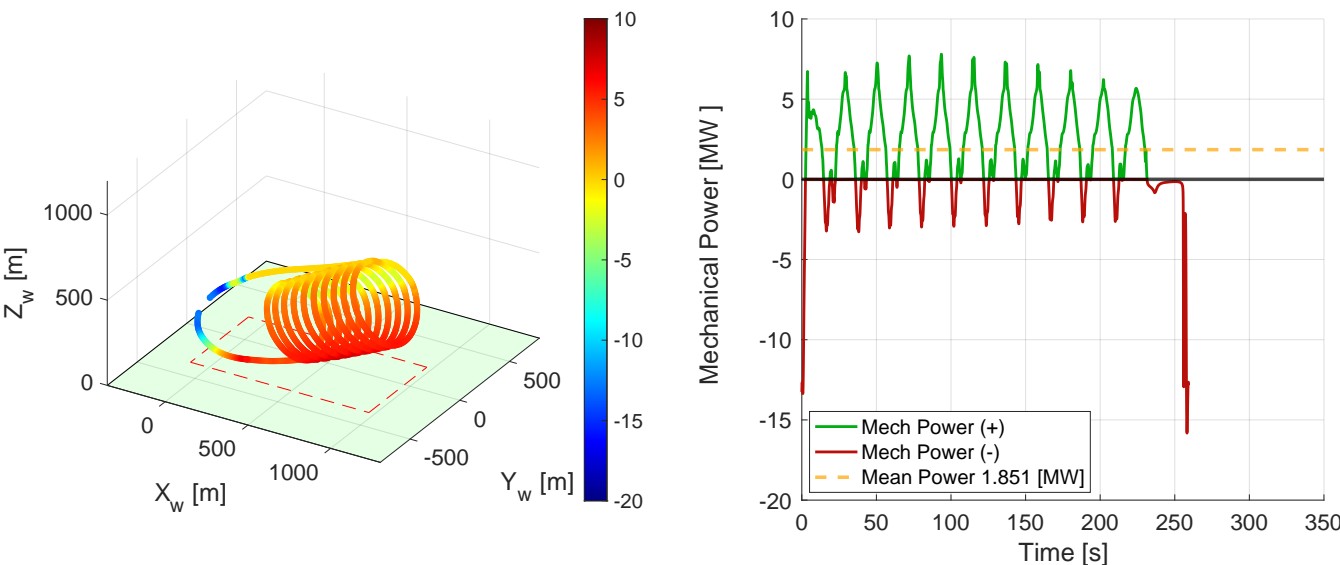

**Figure 11.** Circular flight path of the most optimised solution set, 15 m/s wind speed. Left is the 3D flight trajectory, which is coloured by mechanical power. Right is the mechanical power with respect to cycle time.

gravitational losses which the kite alone cannot compensate for. The cycle average is at 1.851 MW, which is much higher than the figure-of-eight flight patterns.

To illustrate the ground area that the system covers, the projected ground area is shown under the flight path. The converged circular trajectory under these conditions projected on the ground covers an area of 0.63 km$^2$ (0.58 km × 1.08 km). This results in a surface area power density of 2.94 MW km$^{-2}$.

Figure 12 presents the velocity and lift force with respect to cycle time for the circle. As expected the velocity and lift force more or less follow the same behaviour.

The maximum velocity is only exceeded at the start of the cycle. The relatively bad transition from retraction to the traction phase causes this spike in velocity and is, therefore, not considered a limiting factor in the validity of the result for this paper. Transition phases are highly complex and not relevant to the scope of this work. Higher velocity is reached at the end of the downward flight, slowing down on the way back to the upper part of the circle.

Another interesting performance metric that can be extracted from these results is the dominant load frequency of the forces. The most predominant force is the lift force, which, if considering only the traction phase, has a frequency of approximately 0.046 Hertz. Compared to the down-loop, this indicates a higher likelihood of fatigue damage over time as the kite will go through more load cycles more quickly.

Figure 13 shows the figure-of-eight down-loop 3-dimensional flight path coloured by the mechanical power, which is shown with respect to time on the right. The projected ground area is shown in the red dashed box on the ground plane.

It can immediately be noted that at this wind speed, almost all of the mechanical power is positive. Out of the three patterns, this delivers a more smooth power output, requiring less power smoothing equipment. At the centre of the figure-of-eight,

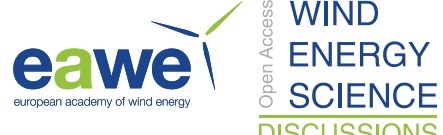

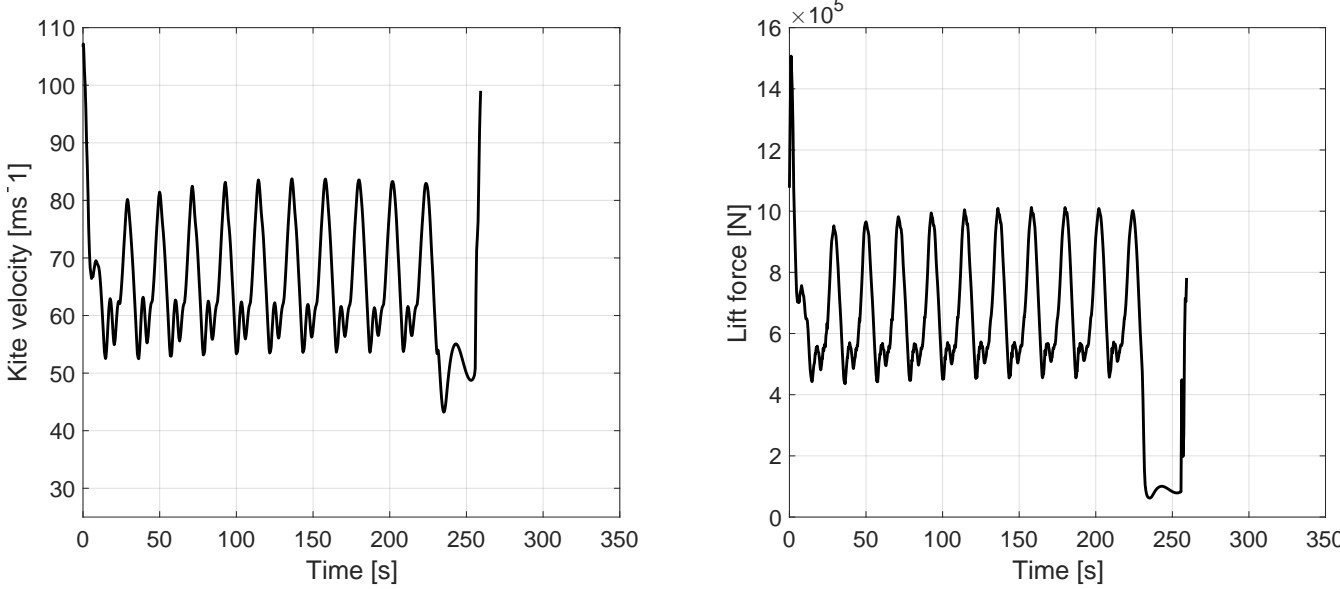

**Figure 12.** Circular flight path of the most optimised solution set, 15 m/s wind speed. Left is the kite velocity with respect to cycle time. Right is the lift force with respect to cycle time.

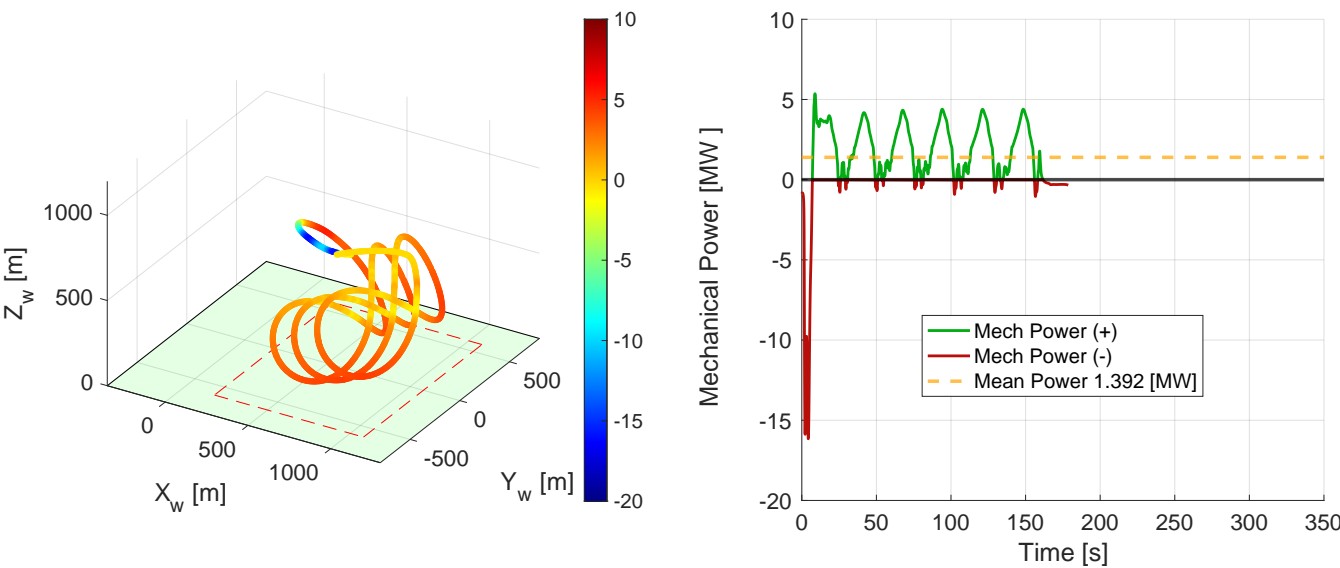

**Figure 13.** Figure-of-eight down-loop flight path of the most optimised solution set, 15 m/s wind speed. Left is the 3D flight trajectory, which is coloured by mechanical power. Right is the mechanical power with respect to cycle time.

the kite is right in the optimal part of the wind window. Using this part to fly up, the forces can use this beneficial wind to





counteract gravity, reducing overall power oscillation. The cycle average is at 1.392 MW, which is lower than the circular path but slightly higher than the up-loop figure-of-eight.

The converged down-loop trajectory under these conditions projected on the ground covers an area of 1.07 km$^2$ (1.19 km × 340   0.90 km). This results in a surface area power density of 1.30 MW km$^{-2}$.

Figure 14 presents the velocity and lift force with respect to cycle time for the down-loop. similarly to the circular path, the velocity and lift force follow the same behaviour.

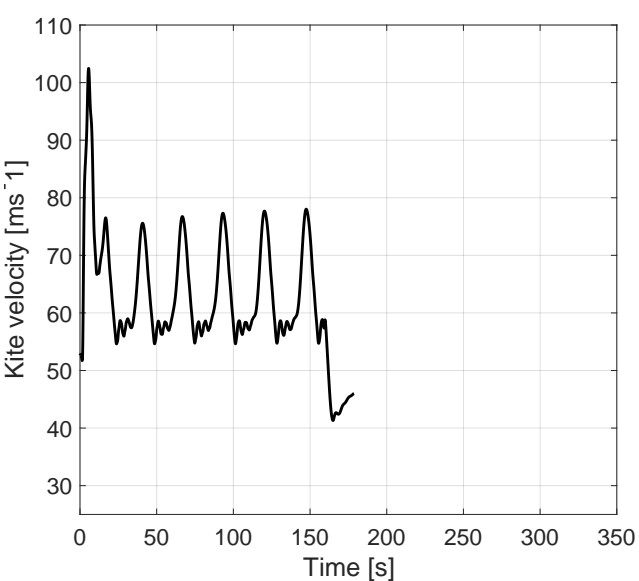

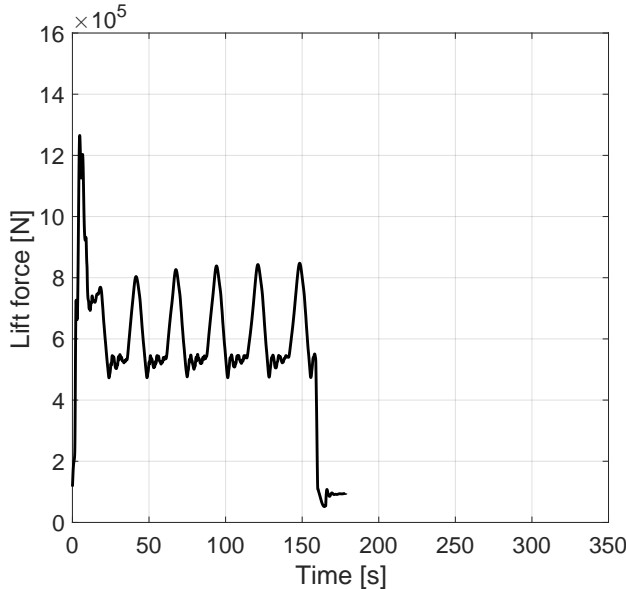

**Figure 14.** Figure-of-eight down-loop flight path of the most optimised solution set, 15 m/s wind speed. Left is the kite velocity with respect to cycle time. Right is the lift force with respect to cycle time.

The maximum velocity is slightly exceeded at the start of the cycle. As this is during the transition phase back into traction, this is ignored. The higher velocity occurs during downward flight at the outer curves, while the kite slows down in the centre, 345   flying up.

The right of Figure 14 shows that the dominant load frequency of the lift force is approximately 0.034 Hertz. Out of the three studied patterns, this is the lowest, having favourable effects on kite lifetime.

Figure 15 shows the figure-of-eight up-loop 3-dimensional flight path coloured by the mechanical power, which is shown with respect to time on the right. The projected ground area is shown in the red dashed box on the ground plane.

It can be noted that at this wind speed, the mechanical power is the most oscillatory of all patterns, requiring more power smoothing than the others. Opposite to the down-loop, the gravitational force needs to be counteracted at the outer edges of the figure-of-eight, where the kite is at the least optimal part of the wind window. Using this part to fly up, the winch has to assist using retraction, increasing overall power oscillation. The cycle average is at 1.344 MW, which is slightly lower than the down-loop figure-of-eight but at worse power quality.





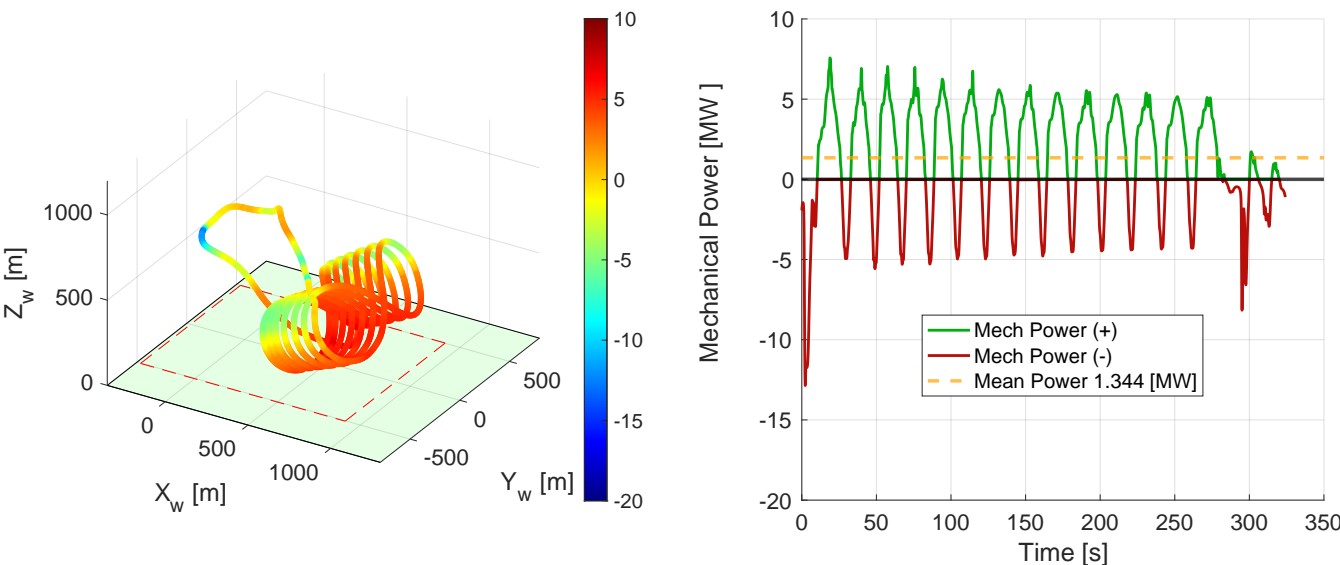

**Figure 15.** Figure-of-eight up-loop flight path of the most optimised solution set, 15 m/s wind speed. Left is the 3D flight trajectory, which is coloured by mechanical power. Right is the mechanical power with respect to cycle time.

The converged up-loop trajectory under these conditions projected on the ground covers an area of 1.24 km$^2$ (1.01 km × 1.24 km). This results in a surface area power density of $1.08 \, \mathrm{MW \, km^{-2}}$.

Figure 16 presents the velocity and lift force with respect to cycle time for the up-loop. similarly to the other paths, the velocity and lift force follow the same behaviour.

The maximum velocity is not exceeded at the start of the cycle; looking at the 3D flight path in Figure 15, this might not

be specific to this flight path. It seems to be a result of the suboptimal transition phase, which just happens to be favourable for velocity. Unlike the down-loop, the higher velocity occurs in the centre, slowing down at the outer edge. Even though the velocity limits of the entire pumping cycle are closer to the average, during the traction phase, the average velocity is higher than for the other patterns.

The right of Figure 16 shows that the dominant load frequency of the lift force is approximately 0.052 Hertz. Out of the

three studied patterns, this is the highest, having unfavourable effects on kite lifetime.

Table 3 gives an overview of the results discussed in this section. Putting the quantities side by side for easy comparison.

## 4   Conclusions

This study systematically evaluated the performance of circular and figure-of-eight flight patterns for fixed-wing AWE systems under consistent operational conditions. Through the development of a Matlab® Simulink®-based controller, the study achieved

effective path tracking and demonstrated clear distinctions between these flight patterns in terms of energy output, power quality, structural load effects and spatial efficiency.



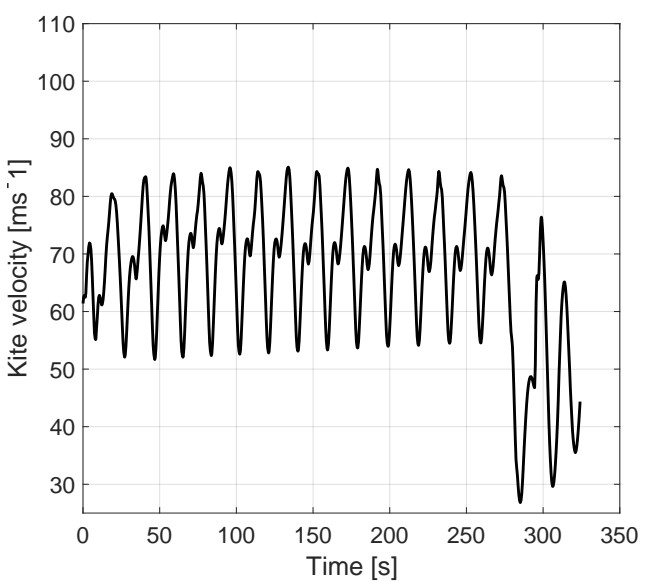

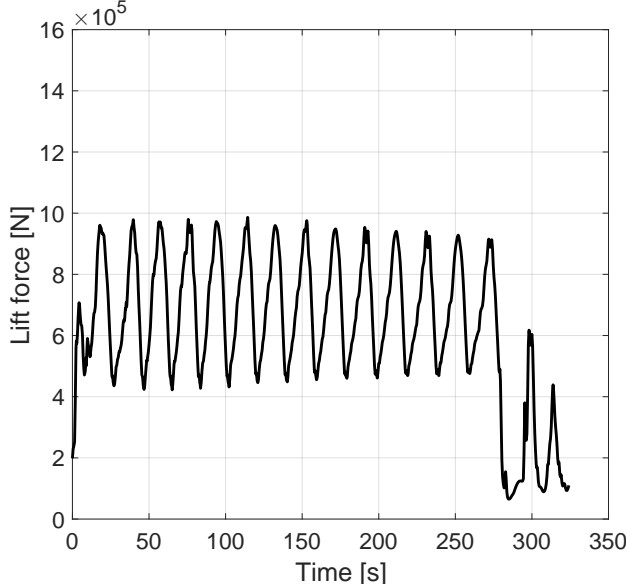

**Figure 16.** Figure-of-eight up-loop flight path of the most optimised solution set, 15 m/s wind speed. Left is the kite velocity with respect to cycle time. Right is the lift force with respect to cycle time.

**Table 3.** Summary of pattern performance of the three studied flight patterns using the MegAWES kite; Circle, figure-of-eight down-loop and up-loop.

|  | Circle | Figure-of-eight down-loop | Figure-of-eight up-loop |
|---|---|---|---|
| Cycle power | 1.85 MW | 1.39 MW | 1.34 MW |
| PAPR | 4.22 | 3.85 | 5.64 |
| Ground area | 0.63 km$^2$ | 1.07 km$^2$ | 1.24 km$^2$ |
| Surface area power density | 2.94 MW km$^{-2}$ | 1.30 MW km$^{-2}$ | 1.08 MW km$^{-2}$ |
| Cycle duration | 259.3 s | 178.6 s | 324.1 s |
| - Traction | 230.3 s (88.8%) | 158.8 s (88.9%) | 279.0 s (86.1%) |
| - Retraction | 29 s (11.2%) | 19.8 s (11.1%) | 45.1 s (13.9%) |
| Maximum kite velocity (traction) | 83.74 m s$^{-1}$ | 78.01 m s$^{-1}$ | 85.07 m s$^{-1}$ |
| Average kite velocity (traction) | 65.86 m s$^{-1}$ | 63.13 m s$^{-1}$ | 70.29 m s$^{-1}$ |
| Dominant load frequency (lift) | 0.046 Hz | 0.034 Hz | 0.052 Hz |

The circular flight pattern yielded the highest cycle-averaged power output of 1.85 MW, outperforming both figure-of-eight patterns in energy capture efficiency. However, it required a smaller operational area, offering a power density of 2.94



$\mathrm{MW\,km^{-2}}$, which is substantially higher than the down-loop and up-loop figure-of-eight paths. These advantages make the circular path attractive for applications prioritizing maximal energy capture within limited ground areas.

Conversely, the figure-of-eight down-loop configuration demonstrated superior power quality, producing lower power oscillations with a peak-to-average power ratio (PAPR) of 3.85, compared to 4.22 for the circular path. This pattern also mitigated gravitational losses more effectively by leveraging optimal wind window positions during ascent. The up-loop variant, exhibiting the lowest cycle power (1.34 MW) and highest PAPR (5.64), suggests it is the least optimal flight path.

Additionally, each pattern displayed different factors which could impact structural fatigue and kite lifetime, driven by variations in load frequency and velocity profiles. The circular pattern showed a load frequency of 0.046 Hz in the traction phase. In contrast, the figure-of-eight down-loop configuration exhibited a lower load frequency of 0.034 Hz, beneficial for mitigating structural fatigue and potentially extending kite life. The up-loop had an even higher frequency and performed the worst.

In conclusion, the choice of flight pattern for AWE systems involves a trade-off between power density, power quality, and operational stability. Circular patterns are preferable for maximizing energy output and spatial efficiency, while the figure-of-eight, especially the down-loop, offers a promising solution for applications where system longevity and smooth power output are critical for grid integration.

Future research should expand the optimization domain, including adjustments to azimuth angles and more refined phase transitions, to further optimize power extraction and load management. Also, other wind speeds should be considered to evaluate if conclusions change throughout the complete wind speed domain.

*Code availability.* The reference design of the kite is available on https://github.com/awegroup/MegAWES in open-access. The simulation environment and parameter inputs used in this work will be made available on the 4TU database.

*Author contributions.* Conceptualisation, D.E., N.R. and R.S.; methodology, D.E., N.R. and R.S.; software, D.E. and N.R.; writing—original draft preparation, D.E.; writing—review and editing, D.E.; supervision, R.S.; funding acquisition, R.S. All authors have read and agreed to the published version of the manuscript.

*Competing interests.* At least one of the (co-)authors is a member of the editorial board of Wind Energy Science.

*Acknowledgements.* This publication is financed by the Dutch Research Council NWO (project "NEON: New Energy and Mobility Outlook for the Netherlands" with number 17628)



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
