# Peer review of "Optimal Flight Pattern Debate for Airborne Wind Energy Systems: Circular or Figure-of-eight?"

_Wind Energy Science, 2024_

## Author Comment (AC2)

**Response to detailed comments of reviewer #2**

Line 21: Missing citation.

**#Response:** This part is rewritten and now includes the missing citation.

**Line 31: "These flight paths have been extensively studied in the context of AWE systems." Please, include the citations of the relevant studies.**

**#Response:** This section is rewritten and does not include that sentence anymore.

**Line 48: Predictable path shapes are not necessarily superior (see, for example, Makani Report No. 1). I suggest rephrasing this sentence.**

**#Response:** The sentence is rephrased, as we indeed did not want convey it was more superior.

Section 1: The following reference is missing from the literature review: G. Licitra et al., Performance assessment of a rigid wing Airborne Wind Energy pumping system (doi: 10.1016/j.energy.2019.02.064).

**#Response:** This study is now included in the introduction section. Also, the differences to this study are mentioned.

**Line 167: It's unclear whether CL and CD refer to the airfoil or the entire aircraft. If they refer to the airfoil, how are 3D effects accounted for? Including a figure showing the aircraft polars (perhaps with and without the tether) would be helpful.**

**#Response:** We agree this paragraph lacks the proper explanation to be clear to the reader. As we think the next section clearly explains the effect of tether drag, and includes a reference to another work, this paper does not need this paragraph and we removed it altogether. The aerodynamic performance of the studied kite is explained in the cited work. This follows the comments of another reviewer requesting the methodology to be more concise.

**Line 173: For aircraft, the lift-to-drag ratio is typically maximized well before stall. Is this also true for the tethered aircraft in this study?**

**#Response:** In general, this is not the case, although we still like to stay away from stall for safety reasons. The tether drag lumped at the kite depends on the tether length, which is variable in the study as the kite flies pumping cycles. However, it is often true that the tether drag is larger than the kite drag. We did rephrase the paragraph, to fix a wrong statement (lift-to-drag-ratio should be the  $C_L^3/C_D^2$  ratio instead, as power is proportional to this ratio instead, already concluded by Loyd). We look at large lift coefficients as the drag coefficient of the tether is generally larger than just the kite, increasing the importance of a high lift coefficient. The optimal angle of attack effectively shifts to a larger value for tethered aircraft than untethered ones. Figure 1 shows this effect. However, we think this figure does not fit within the content of the paper.

Figure 1:  $C_L^3/C_D^2$  with different tether lengths for the MegAWES kite.

**Line 177: If the polar plots were given, it would be clearer whether a 4-degree angle of attack is close to stall. What CL corresponds to 4 degrees?**

**#Response:** As is mentioned above, the aerodynamic performance of the Kite is already published in the cited paper. In order to stay concise, this is chosen as the best compromise of including and referring to data. In that work, it is explained how the kite is kept in the linear region of the lift curve and how the maximum angle of attack for this region is determined. The CL corresponding to a 4-degree angle of attack would be 1.8, this is now added to the section as well. The aerodynamic method employed to obtain the lift curve had to stay in the linear region to ensure validity, as stall could not be predicted with the 3D panel method called Apame. The low 4 degrees angle of attack is somewhat misleading, though, as the wing has a structural twist of 5 degrees at the root. This makes the equivalent operational maximum angle of attack closer to 9 degrees.

**Line 200: What is the kite position? Its center of mass?**

**#Response:** This is rephrased; the position is indeed the centre of mass.

**Line 213: Again, which reference point on the wing is used?**

**#Response:** It is not fully clear to the authors what is meant here as this sentence does not refer to the wing. It is, however, clarified where the winch is positioned.

**Line 222: What is the "equivalent wind speed"?**

**Response:** It means the wind speed in the direction of the tether. However, this quantity is not directly used in the framework; it is only part of the derivation. Therefore, we decided to rewrite the equation to simplify the definition.**

**Line 225: Please provide a brief definition of $f^*$ .**

**#Response:** A sentence is added to define the reel-out factor f and its optimum  $f^*$ .

**Line 233: Why was this optimization algorithm chosen? Please justify this choice. Also, the reference for the algorithm links to a presentation rather than a peer-reviewed paper.**

**#Response:** The reference is changed to the actual published and peer-reviewed book chapter on the CMAES method. A justification is now provided in the text. This study built upon previous research that

had already employed the CMA-ES method successfully. Other methods, like the Matlab built-in FMINCON and GA method, were much less successful in providing a stable framework.

Line 243: I have concerns about the objective function. Why not use average power as the sole objective function and treat the other factors as constraints? If the chosen optimization algorithm does not support constraints (which I suspect), why not use a different algorithm? The choice of the objective function directly impacts the results, making this a multi-objective optimization problem where the five objectives have completely arbitrary weights. Please justify the weights selection. How could these weights be applied to other systems?

**#Response:** As this paper is not about optimisation, but rather a method to numerically explore parameter combinations, the weights are only there to work the optimiser towards a solution with zero penalty. As mentioned in the general response, the objective might have been misunderstood. If the mentioned values are not exceeded the penalties are zero. Therefore, a solution without penalties is what is aimed for and, thus, a constrained single objective optimisation. However, the applied weights were carefully selected to stabilise the process and fulfil the constraints. This exploration method proved to be the most successful in achieving this result, compared to the built-in Matlab GA and gradient-based methods. Especially, as we designed the framework to work with different kite designs where the initial condition is unknown. We have adapted the text to clarify this, emphasizing more this is not about finding an optimal flight path but rather to arrive at a path consistently obtained using the same objective to make the comparison more fair. We do acknowledge that changing the optimisation method and/or weight would maybe converge to a different, if not better solution, but this would require a comparison of different methods, tuning each and every one of them to work properly, which to the authors seems excessive and beyond the scope of this work. A future study could definitely focus on just the optimisation part, comparing the behaviour and convergence of different methods and their impact on the optimal trajectory for each pattern. The developed framework could function as the starting point of such a research project.

**Additionally, some of these objectives may be insignificant (based on the figures, I suspect pVk = 0 and pFt = 0), while others may have large values (PAPR values in Table 3 exceed 2.5). However, these details are not explicitly provided in the results and should be included.**

**#Response:** This is mostly answered in the previous response. We do not look at multiple objectives. Indeed, the PAPR values never manage to go below the 2.5, which is now also mentioned in the paper. There could be several reasons for this, one of which might be that we are stuck in a local optimum that in the current conditions does not manage to reduce the PAPR to 2.5, but at least tries to minimise it. Another reason could be that it is a limit of the longitudinal controller. In the paper we refer to another work, which explored minimising the power peaks by an additional tether force controller.

**Section 2.5: The comparison criteria should align with the objectives. Two of the criteria are included in the objective function, but one is not—please elaborate on the reasons.**

**#Response:** In general this would be true. However, this work intended to compare these criteria based on comparable flight paths, which are obtained by optimising the power.

Line 276: Why was 15 m/s chosen? Does this represent below-rated condition? Which wind shear is considered for the study? I cannot find this information.

**#Response:** The wind shear used was indeed not mentioned, this is added to the subsection system parameters. We use the measurement data from the Ijmuiden offshore measurement mast and after the maximum wind speed is reached the wind speed remains constant (at 250m altitude). The 15 m/s is indeed a below-rated condition. This allows the operation to be at maximum production capability as the kite should perform less optimally after rated power, which is beyond the scope of this work. Even though a rated power was never chosen for the MegAWES kite, during winch sizing (separate work), a 3MW limit was set, corresponding to about 20 m/s wind at operational altitude. The 15 m/s wind is the maximum wind speed in the simulation and thus constant from 250 meters altitude. Effectively, this means the tether only sees the wind shear, the kite experiences a constant wind speed of 15 m/s. Even though wind shear would add some variability in velocity throughout the pattern, we decided a study using a different wind shear model would be good for future analysis.

Line 285: Please provide citations for these statements. A low turning radius causes the aircraft to bank inward, reducing its projected area relative to the wind direction. As discussed in Makani's first report, this is one reason for avoiding low turning radii. Additionally, low turning radii experience greater wind speed reduction, though this effect does not appear to be considered in this study.

**#Response:** Indeed the first Makani report discusses the effect of banking on tether tension and thus finds the optimal path radius based on maximising power. This is included, as the forces and orientation of the kite influence the dynamics and thus time stepping. Which affects the force balance of the overall system and finally affects the power. We added this statement to the discussion. The explanation does not come from a reference, but is rather derived from the definition of centripetal acceleration and force, which are also mentioned in the Makani report. In order to make the patterns comparable, the cross-track error was set to a maximum of 75m. This way, we ensure the kite is always close to the prescribed path, and therefore, we can use the prescribed path as a comparative measure rather than the actual flown trajectory. A smaller radius and similar velocity would increase the centripetal force. To maintain control and prevent drifting outwards, either the velocity should increase or the aerodynamic performance. However, searching for power optimal solutions and assuming the wind speed is below rated, one can assume the kite operates around its aerodynamic maximum capabilities, leaving only the velocity to compensate. This is indirectly changed throughout the course of the parameter exploration as velocity cannot be directly controlled. That being said, the increased velocity also increases the aerodynamic forces. The upwind wind speed reduction for low turning radii is indeed not included, as there is no wake model. We believe a wake model would be too detailed for this work. The effect would further strengthen the statement that there is a limit on how small the turning radius would become.

**Figures 8, 9, 10: These figures might be more appropriate for the appendix. Since this paper focuses on optimization results rather than the optimization process itself.**

**#Response:** These three figures provide information on the relation between the power and the different pattern sizes. We do acknowledge that the paper might have focused too much on the word optimisation rather than constrained design space exploration towards a comparable path. Therefore, the images are modified to show the relations between the pattern dimension and performance criteria more clearly.

---

## Author Comment (AC3)

**Response to detailed comments of reviewer #3**

**Line 21-22: missing citation.**
**Response:** Correct citations are added**

**Line 103: a dynamic system -> a dynamical system**
**Response:** Rewritten to the suggested words.**

**Line 104: I don't know much about L0 and L1 controller. Based on what's written, L1 is the distance between the kite and the target (i.e., the error). If so, how could L1 be an adjustable parameter?**
**Response:** The L1 is not calculated as an error, it is the distance to the path where the target should be defined. Basically, it defines how far the controller looks ahead to determine the target. We added a sentence to the controller section to clarify this: "This distance is often used as a control parameter to ensure that the system moves toward the desired trajectory by looking either closer or further ahead to determine an optimal target position to track."**

**End of line 140: the -> to**
**Response:** Rewritten to the suggested word.**

**Equation 3: are the phi_tau terms the roll angles (desired/actual) on the Earth axis or the tangent axis? I assume it's the latter. Please specify.**
**Response:** We believe the last sentence before equation 3, together with the tau subscript should clarify it refers to the tangential plane: "Finally, by considering all forces acting on the kite, the desired roll angle $\phi_{\tau,\text{des}}$, relative to the tangent plane, is determined by Equation (3)."**

**Line 157: is lift F_L taken directly from the aerodynamic model, or is there a way to measure it in-flight in real life?**
**Response:** Lift is generally impossible to measure in real life. In this simulation framework, there is no mismatch between the aerodynamic model and kite force. In real life, one would need to be able to deal with this model mismatch, as the controller would only know a predicted lift instead of the real lift. Several methods could be employed to reduce the risk of that becoming an issue. Either by improving the aerodynamic model to better predict real forces or by changing to a controller which can better handle the mismatch. This work aimed for a simple tunable controller for numerical studies only. For a real kite controller, this might come with several problems that need to be addressed.**

**Line 193: 'this thesis' – please update.**
**Response:** Rewritten the sentence.**

**Line 217: please specify whether this is the tether force on the winch or the aircraft body, or are they the same?**
**Response:** We rephrased the text to clarify this. It is the ground tether force, as this, in the end, is what is converted into electricity, so this is what the controller tries to track. The tether force at the kite, is quite similar to the ground during the traction phase when there is little to no tether sag, during reel-in when the**

tether sag is much larger the two differ significantly. The tether model we are using captures this difference.

**Line 256: I guess the word 'using' shouldn't be there.**
**Response:** We rephrased the sentence.**

**Lien 286: become -> be**
**Response:** We rephrased the word.**

**Line 288: the maximum -> the maximum velocity**
**Response:** The word 'velocity' is added.**

**Line 307 and others: the authors use the term 'most optimal'. I don't think that's appropriate because an optimal point is the best one, unless we are referring to local and global optima.**
**Response:** That is indeed what we refer to. We can never be sure we have a global optimum for such a large non-linear framework, even under specific conditions.**

**Line 315-316: "the winch needs to retract a bit to insert energy into the system to get it to the top of the circle." Is this common in airborne wind? I would have thought that if one needs to add more energy to the kite via the winch, it means the target trajectory is too challenging and the kite should fly at a lower height.**
**Response:** It is a common thing we see in simulations for heavy kites. However it is an undesirable phenomenon. As is also concluded in this paper, it is bad for the power quality. This is not generally solved by flying lower, however, flying lower could mean a lower elevation angle, and this decreases the cosine losses having a positive effect on the achievable aerodynamic force. If with lower, a shorter tether length is meant, this could reduce the tether drag and increase system performance. But, flying lower can also come with other problems like minimum required altitude or other flight domain size requirements. In this case the elevation angle is already quite low, so we do not expect a large difference with a lower altitude. It must also be noted, that this kite is designed using a single element airfoil with underperforming aerodynamic characteristics due to the limited access to advanced aerodynamic solvers at the time. Therefore, the kite does not perform too well compared to its mass, causing these gravity-induced problems. As we keep the kite constant throughout the analysis, this effect is for all patterns the same, so it would still make up for a fair comparison. We as authors have also worked with a different controller on this kite, which showed that this phenomenon could be minimised using alternative (more complex) pitch controller strategies, but was only tested for the traction phase of an up-loop figure-of-eight pattern (DOI: 10.1088/1742-6596/2767/7/072019, also in the list of references and referred to in Section 2.1.3).**

**Line 330-331: "Compared to the down-loop, this indicates a higher likelihood of fatigue damage over time as the kite will go through more load cycles more quickly." This is unclear. Does the figure-of-eight frequency cover 1 orbit on 1 side or 2 orbits (1 on each side)? If it is 2 orbits (1 on each side), then the figure of eight is expected to have a lower frequency than circular flight, but each figure of eight cycle stresses the aircraft (roughly) twice as much a circle cycle because we are doing 2 circles (1 on each side). The comments on aircraft stressing might need some further thinking.**

**#Response:** This is only true if the pattern is flown in the same time. We do not look at a load cycle as the flown pattern, even though this is generally the case for a circle, and half that of a figure of eight. We look at the cyclic lift force compared to time independent of the flown pattern. The given frequency, therefore gives a simplified early indication of the amount of cycles in a life time. We did rephrase the discussion to better explain what is meant, and we added a sentence about the force amplitude, which supports the given argument even more. However, it is also stated as an additional result, and must be interpreted with caution, especially the frequency. The relative amplitude can be expected to be more consistent with the pattern choice.

---

## Author Comment (AC4)

**Response to detailed comments of reviewer #4**

line 21: Citation seems to be noted, but not given
**Response: The citation is now added where it was supposed to be.**

**Figure 2: The variables l and s are not defined**

**#Response:** We added the definition in the caption, as the image should be readable without the text. Of course, a more detailed description of the parameters can be found in the accompanying text.

**Line 129: given the focus of this article on paths, I think some further description on what Lissajous, Lemniscate are.**

**#Response:** We have added some additional explanations on the patterns in this section.

Line 150: is referenced before 4 or 5, I can see the logic in the chosen ordering of figures, in that this out-of-order reference to 6 is to better illustrate a detail. However, since this section already discussing geometry, for example equation 3 refers to different components of gravity along different axis. I would suggest that figures 5 and 6 are described early to establish the kinematic convention at the start.

**#Response:** We have moved this subsection to be the first in section 2. Therefore, a combined version of figures 5 and 6 is now presented as figure 3.

Figure 4: It's assumed that the same structure is used for both longitudinal and lateral control. Could be made more specific. Labels are a bit vague, could be made more specific. Instead of angle, pitch/roll angle, instead of rate, angular of pitch/roll rotation, instead of deflection, elevator/aeleron deflection

**#Response:** We have edited the figure, to incorporate your suggested terms, to make the labels less vague.

**Equation 8: PAPR not defined, cross-track error not described, P\_mech is intuitive, but again, not described. Maybe it's described later, but since the multi-objective problem is given here, all these different objectives should be elaborated here.**

**#Response:** We have tried to make it more clear that it is not a multi-objective problem. The result is chosen based on the single objective of average power. The solution is then compared using multiple criteria explained elsewhere. The given penalties are needed to obtain comparable results and to make sure hard limits are not exceeded in a simulation, but the final solutions have zero penalties (except for PAPR), making it just a single-objective problem with a minimized PAPR. Imposing the limits within simulink proved to be a too harsch criteria for when initial conditions are unknown for a given kite design. In these cases the optimiser never really started. Using penalties as a guide proved to be a successful strategy for obtaining solutions that converged and did not exceed the limits. Additionally, the entire section is modified, and now it also has the terms defined. We also rewrote the equation so it is more clearly shown, there could be a zero penalty.

**Section 2.5, the objectives of comparison are given, it could be helpful to give the mathematical definition of these metrics that were used by authors. One could imagine alternative equations for the same metrics.**

**#Response:** We have added the mathematical definitions to section 2.5.

**Line 303, refers to figure 9, but seems to talk about figure 10 results.**

**#Response:** We fixed this mistake, it was indeed referring to the wrong figure.

Throughout the paper, the author refers to the frequency of lift variations as an indication that fatigue. However, the magnitude of the force oscilations is also an important factor. Looking at this parameter, further supports the authors conclusion. This should be mentioned to give more weight to their arguments. They could even estimate a damage rate directly from the data to give a more solid conclusions on fatigue.

**#Response:** This is indeed true, we rephrased the discussion and included the amplitude as well. However, as fatigue was initially not the focus of the comparison, we think a qualitative analysis on the fatigue is enough to show that besides power related metrics, the pattern could also have an effect on the structure. In the conclusion section we also rephrased the paragraph on fatigue results.